# Rates and risk factors for antepartum and intrapartum stillbirths in 20 secondary hospitals in Imo state, Nigeria: A hospital-based case control study

**Uchenna Gwacham-Anisiobi**[1]*, **Charles Opondo**[2], **Tuck Seng Cheng**[1], **Jennifer J. Kurinczuk**[1], **Geoffrey Anyaegbu**[3], **Manisha Nair**[1]

**1** Nuffield Department of Population Health, National Perinatal Epidemiology Unit, University of Oxford, Oxford, United Kingdom, **2** Department of Medical Statistics, London School of Hygiene and Tropical Medicine, London, United Kingdom, **3** Health Strategy and Delivery Foundation, Abuja, Nigeria

* uchenna.gwacham-anisiobi@balliol.ox.ac.uk

## Abstract

Despite Nigeria's stillbirth rate reducing from 28.6 to 22.5 per 1,000 births from 2000–2021, progress trails comparable indicators and regional variations persist. We assessed stillbirth incidences and associated risk factors in 20 secondary hospitals in Imo state, to generate essential local evidence to inform policymaking to reduce mortality. The total numbers of births and their outcomes were determined through hospital maternity registers. An unmatched case-control study was conducted. We collected retrospective data about 157 antepartum and 193 intrapartum stillbirths, and from 381 livebirths (controls). Potential risk factors were categorised into sociodemographic, obstetric and maternity care and biological determinants using a theoretical framework. Independent multivariable logistic regression models were used to investigate the association of risk factors with each stillbirth type. The overall stillbirth rate was 38 per 1,000 total births. The rate of antepartum and intrapartum stillbirths were 16 and 19 per 1,000 respectively. The risk factors independently associated with antepartum stillbirths were nulliparity (adjusted odds ratio (aOR) 1.87, 95%CI 1.04–3.36); preterm birth (aOR 14.29, 95%CI 6.31–32.38); being referred from another facility (aOR 3.75, 95%CI 1.96–7.17); unbooked pregnancy (aOR 2.58, 95%CI 1.37–4.85); and obstetric complications (aOR 4.04, 95%CI 2.35–6.94). For intrapartum stillbirths, associated factors were preterm birth (aOR 11.28, 95%CI 4.66–27.24); referral (aOR 2.50, 95%CI 1.19–5.24); not using a partogram (aOR 2.92, 95%CI 1.23–6.95) and obstetric complications (aOR 10.71, 95%CI 5.92–19.37). The findings highlight specific risk factors associated with antepartum and intrapartum stillbirths, shedding light on potential areas for targeted interventions.

## Introduction

Globally, approximately 1.9 million stillbirths occurred in 2021, with 98% of these in low- and middle-income countries [1]. Sub-Saharan Africa and South Asia accounted for 47% and 32%

**Data availability statement:** The data relevant to this study has been included in the article or included as Supporting Information.

**Funding:** This study is part of UGA's DPhil research which is jointly funded by Nuffield Department of Population Health, Balliol College and the Clarendon Fund. The study funders had no role in study design, data collection, data analysis, data interpretation or in writing of this manuscript.

**Competing interests:** The authors have declared that no competing interests exist.

of global stillbirths respectively [1]. The death of a baby after 28 weeks of pregnancy, but before or during birth is defined by the World Health Organisation (WHO) as a stillbirth. Based on the time of occurrence, stillbirths are classified as antepartum stillbirth (ASB) (fetal death before the onset of labour) and intrapartum stillbirth (ISB) (fetal death during labour or birth).

There were 182,307 stillbirths in Nigeria in 2021 accounting for 10% of all stillbirths worldwide [2]. Although there has been a decrease in the rate of stillbirths in Nigeria from 28.6 to 22.5 per 1,000 total births between 2000 and 2021, progress is slow with an annual reduction rate of 2.3%, compared with 2.9% and 4.3% for neonatal and under 5 deaths respectively [2,3]. Over 45% of stillbirths worldwide occurred during labour and birth [1], but in West and Central Africa, this was 52% [1]. Within Nigeria, there are regional disparities with a greater burden estimated to be in the northern regions [4]. Achieving the Every Newborn Action Plan's target of reducing stillbirths to ≤12 per 1000 total births [5] remains a challenge in Nigeria.

The existing studies of stillbirths in South-eastern Nigeria were either single facility or multicentre cross-sectional tertiary facility-based studies [6–9]. Two studies conducted in tertiary facilities in Imo state over ten years ago reported rates of 60 stillbirths per total 1000 births (67% ASB and 33% ISB) [7] and 180 stillbirths per 1000 births (75% ASB and 25% ISB) [6], respectively. However, the tertiary facilities receive complex referrals and as such findings are skewed by high-risk pregnancies. Secondary facilities provide the majority of the comprehensive emergency obstetric care in the state [10] and therefore it is important to generate vital evidence from these facilities to guide policymaking and efforts to reduce stillbirths in the region. The objectives of our study were to: (i) estimate the incidence of overall and types of stillbirths in Imo State; (ii) investigate the risk factors independently associated with antepartum and intrapartum stillbirths.

## Methods

### Study design

A case-control study, using data from medical records of women who gave birth in 20 secondary facilities in Imo state, was conducted. Due to political unrest in the state, facility recruitment was restricted to areas considered 'safe' by local security intelligence agencies. Cases were women reported as having stillbirths in their medical records and controls were women who had a livebirth between January 1, 2020 and July 31, 2022. The controls were selected from maternity registers as the livebirth preceding each case for convenience and to minimise selection bias. The cases and controls were not otherwise matched.

### Study setting

Imo State has an estimated population of 5,167,722 and an estimated total fertility rate of 4.5 which is below the national average of 5.3 [4]. The majority of its inhabitants are from the Igbo ethnic group, and the predominant religion is Christianity. Around 91% of women of reproductive age have secondary education or higher [4]. Additionally, 92% of households have access to clean water, 70% have electricity, there is a 12.5% poverty rate [11] and 26.1% unemployment rate [10] which make healthcare affordability a challenge. In 2019, 68.6% of households in the state faced catastrophic health expenses, as per World Bank definition, with only 1.7% having health insurance coverage [11]. There are 602 Secondary facilities (3.2% government and 96.8% non-government run) compared with two government tertiary facilities in the state [10]. Over 70% of women opt for private maternity services [4], underscoring the need for research into stillbirth occurrences, types, and risk factors in non-government secondary facilities, which provide the majority of Comprehensive Emergency Obstetric Care (CEmOC) services and constitute over 95% of secondary maternity facilities in the state [10].

## Study sample

Hospital records of all women who gave birth at 28 weeks or later between January 1, 2020 and July 31, 2022 formed the sampling frame in the 20 selected hospitals. Sample sizes were computed for independently assessing ASB and ISB to enable detection of odds ratios of ≥2.0 for risk factors with 5% to 25% prevalence respectively (S1 Table). On this basis, using the Kelsey equation [12], a sample size of 173 cases each of ASB and ISB, and 346 controls (1:2 ratio) adjusted for clustering was required (S2 Table). For efficiency the same control group was used in the analysis for both types of stillbirths.

## Data sources

There were two sources of data for the study. Information about the total number of births and the outcomes in the study period was extracted from the maternity registers using data collection forms. Following the identification of cases and controls from maternity registers, information about ASB and ISB (cases) and controls, was collected from the women's medical case folders by trained midwives and nurses using pre-tested forms and supervised by co-authors UGA and GA.

## Variable choice and extraction

For consistency, stillbirth classification into ASB and ISB was based on the classification in the maternal records. Potential risk factors for ASB and ISB in Nigeria were identified through a systematic search of Embase and MEDLINE. These potential risk factors were categorised into sociodemographic, obstetric and maternity care and biological factors adapting the Mosley and Chen (1984) theoretical framework [13](Fig 1). Only variables in Fig 1 which could be abstracted from patient medical records were included in this study.

## Statistical analysis

The incidence of overall and types of stillbirths were calculated for the total period of data collection and by facility type (public, mission, private) using counts of stillbirths and livebirths.

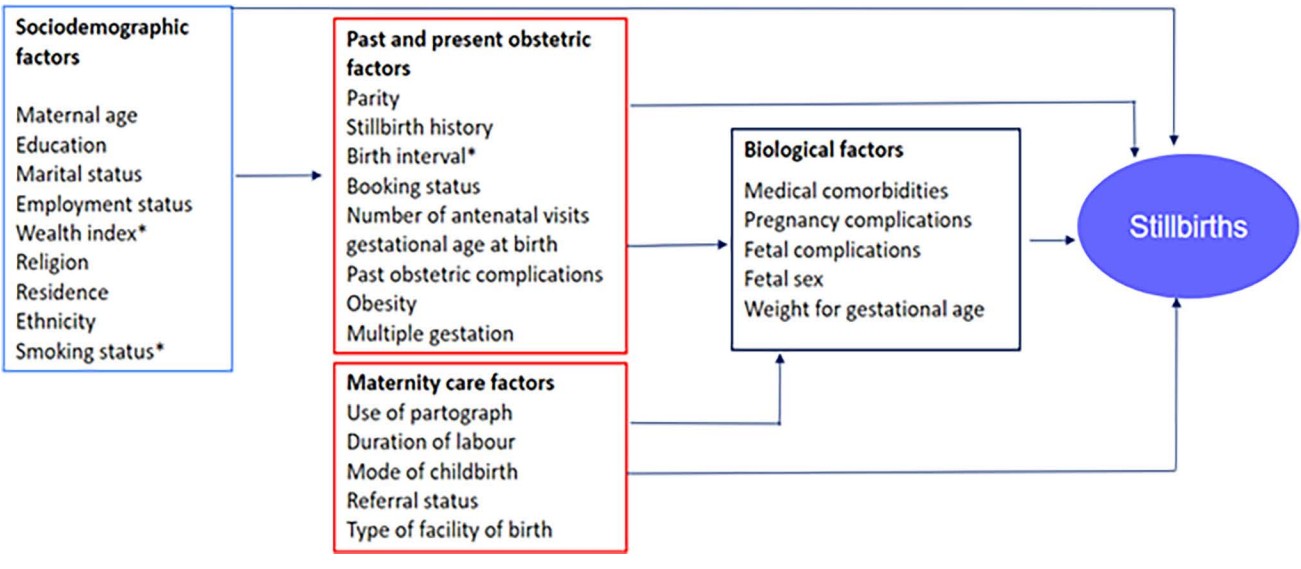

**Fig 1. Theoretical framework.**

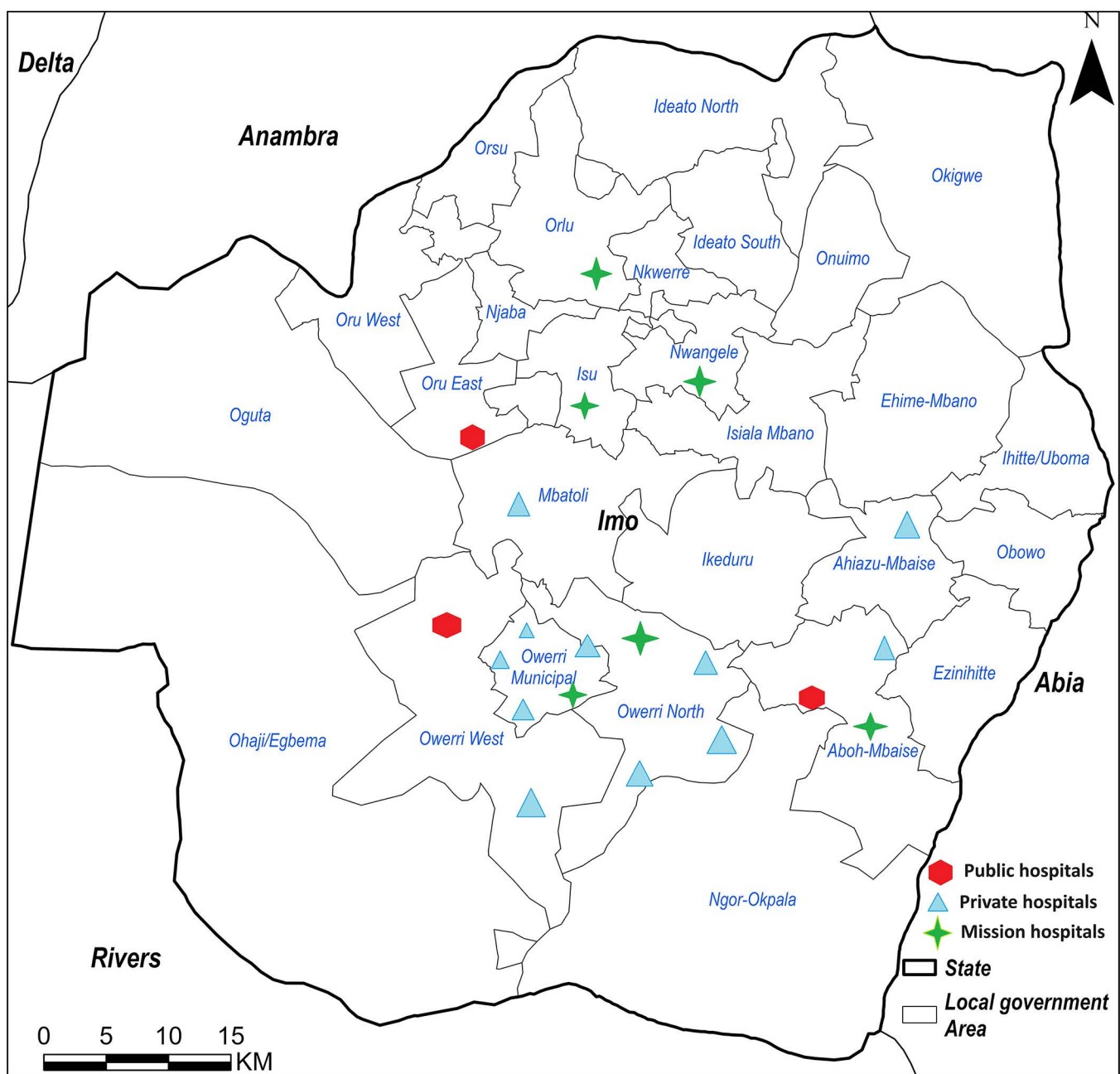

**Fig 2. Geographical distribution of health facilities included in the study overlaid on the local administrative boundaries in Imo state Nigeria.** Source: The map was plotted by the author in ArcGIS Pro V 3.3 (ESRI, Redlands, CA, USA) based on boundaries from GRID3 under a CC BY 4.0 License (https://data.grid3.org/datasets/GRID3::grid3-nga-operational-lga-boundaries/about).

Quantitative variables were summarised using mean or median, based on their distribution, while categorical variables were summarised using frequencies and percentages. The characteristics of cases (antepartum and intrapartum) and controls were compared. Variables that had more than 15% missing information were not included in the primary models.

Univariable logistic regression analysis was used to assess the individual associations of potential risk factors with ASB and ISB separately. Variables with a p-value < 0.10 in the

univariable analysis were included in the sequential multivariable logistic regression models; the order of inclusion of the variables was guided by the theoretical framework. For continuous variables such as maternal age and gestational age, tests for departure from linearity and trend were conducted. We planned to analyse the association of the mode of childbirth (SVD, assisted vaginal birth, elective and emergency CS) with intrapartum stillbirth. However, the numbers were too small for a robust analysis and including these them in a regression model would have led to unreliable estimates and potentially inflated standard errors. Therefore, we decided to group all vaginal births together and all caesarean births together to maintain the statistical validity of our analysis. If the effect size of the OR for an independent variable changed substantially after adjustment, we explored the influence of confounding by other factors. We also assessed multicollinearity using Spearman rank correlation, $r_s > 0.80$ or $<-0.80$) and calculated variance inflation factors (VIF). The associations between risk factors and stillbirth were presented using adjusted odds ratios (aORs) with 95% confidence intervals (CIs).

For the final models for ASB and ISB, an assessment was conducted to examine the patterns and quantities of missing data for each variable included. Although missing data could theoretically be linked to other independent variables, missingness was not thought to be associated with the outcome, hence it was presumed that information were missing at random. Therefore, the primary models were complete-case analyses. Multiple imputation by chained equations was used to address missingness, and estimates from this model were compared with the complete-case analysis.

To assess how well the models distinguish between the risk factor groups (sociodemographic, obstetric and maternity care and biological factors), we compared the area under the receiver-operating characteristic (AUROC) curves for the models. Population Attributable fractions (PAF) were calculated for risk factors that were found to be significantly associated with stillbirth. STATA version 17 was used for analysis.

Ethics approval was obtained from Federal Medical Centre, Owerri (FMC/OW/HREC/VOL.II/68), Imo State Specialist Hospital and Oxford Tropical Research Ethics Committees (OxTREC Ref: 535–22). Both ethics committees waived the requirement for informed consent. Only UGA and the research midwives in each facility had access to the identifiable medical records at the point of data collection in each facility. Each participant's data were anonymised at point of collection making it impossible to trace data back to participants after daily validation. The rest of the research team accessed only fully anonymised data for the analysis. There was no direct contact with participants for this study.

### Role of the funding source

The study funders had no role in study design, data collection, data analysis, data interpretation or in writing of this manuscript. UGA, GA and TSC had access to the data in the study and all authors accept responsibility for the decision to submit for publication.

## Results

Twenty hospitals located in 10 of the 27 Local Government Areas in two geopolitical zones in the state (Fig 2 and S3 Table) contributed to the study. These included 12 private, five mission and three government hospitals.

There were 11,278 births in the 20 hospitals between January 1, 2020 and July 31, 2022. Of these, 10,898 were livebirths and 432 were stillbirths, resulting in a stillbirth rate of 38 per 1,000 total births (Table 1); 42% were antepartum, while 49.5% and 8.3% were intrapartum and unclassified stillbirths respectively. The term 'unclassified' in this study denotes instances where the stillbirth type is undocumented in maternal register and either the medical records were unavailable for review or had insufficient information to decipher the type of stillbirth.

**Table 1. Stillbirth rate per 1,000 total births by facility types.**

| Facility type | Total births N | Stillbirths n | Overall stillbirth rate per total 1000 births | Antepartum stillbirth n (rate per 1000 total births) | Intrapartum stillbirths n (rate per 1000 total births) | Unclassified stillbirth n (rate per 1000 total births) |
|---|---|---|---|---|---|---|
| Public hospitals | 2,460 | 110 | 45 | 60 (24) | 47 (19) | 3 (0.4) |
| Mission hospitals | 2,948 | 159 | 54 | 56(19) | 87 (30) | 16 (5) |
| Private hospitals | 5,870 | 163 | 28 | 66 (11) | 80 (14) | 17 (3) |
| Total | 11,278 | 432 | 38 | 182 (16) | 214 (19) | 36 (3) |

Total stillbirths– 432: 42.1% (182) antepartum, 49.5% (214) Intrapartum, 8.3% (36) unclassified.

This definition differs from its use in Relevant Condition at Death (ReCoDe) classification, which is a system used to classify the underlying cause of death in perinatal death and takes into account both maternal and fetal conditions to determine the primary cause of death [14]. The rate of overall stillbirths was highest in Mission hospitals (54 per 1,000 births), while private hospitals had the lowest rate (28 per 1,000 births). ASB was highest in public hospitals (24 per 1000 births) while ISB was highest in Mission hospitals (30 per 1,000 births).

Following the exclusion of missing records, unclassified stillbirths and twin gestations, 731 singleton records were included in the risk factor analysis. Within this sample, there were 157 ASB, 193 ISB, and 381 livebirths (S1 Fig).

The characteristics of women included in each group are reported in Table 2. The mean maternal age of the ASB, ISB and control groups were 29.7, 29.9 and 28.7 years respectively. Over 70% of all cases and control had no prior history of stillbirths. About half of the women who had ASB were not booked for pregnancy care in any hospital, and 53.4% of pregnant women with ISB were booked in the facility of childbirth, compared with 81.1% of controls. About 69% of controls had four or more antenatal care visits compared with 36% and 48% for women with ASB and ISB. While only 4.2% of controls had a preterm birth, about a third of the cases (women with ASB 35%, ISB 27.5%) had preterm childbirth. When indicated, a partogram was only used during labour for 19.4% of women with livebirths compared with 8.8% of women with ISB. Nearly half of the women with ASB were referred from other facilities compared with 38.9% and 12.9% for women with ISB and livebirths, respectively.

As over 90% of women in the sample were of Igbo ethnicity and identified as Christians, these variables were excluded from the analysis due to their limited variability across the outcome groups. Additionally, maternal education, employment status, and fetal weight for gestational age were omitted from subsequent analysis due to each having more than 40% missing data.

Results of the crude associations are presented in S4 Table. After adjusting for maternal age and obstetric and maternity care factors, maternal age (aOR 1.05, 95%CI 1.00–1.11); being nulliparous (aOR 2.02, 95%CI 1.15–3.56) compared with having 1–3 previous childbirths; preterm birth compared with term birth (aOR 13.61, 95%CI 6.24–29.65); being referred from another facility (aOR 3.75, 95%CI 1.96–7.17); and being unbooked compared with booked and having four or more antenatal visits (aOR 2.58, 95%CI 1.37–4.85) remained significantly associated with ASB (Table 3).

After adjusting for biological factors, maternal age was no longer significant, and the association was slightly attenuated for the obstetric and maternity care factors (see Table 3). Having any obstetric complication, was associated with a four-fold increase in the odds of ASB (aOR 4.04, 95%CI 2.35–6.94), but pre-existing medical comorbidities was not significantly associated with the outcome (Table 3). There was no indication of multicollinearity in the model (mean VIF—1.23, maximum VIF—1.65; $r_s$ = -0.6 to 0.4).

**Table 2. Characteristics of cases and controls.**

| | Cases<br>Antepartum stillbirth (N = 157) n (%) | Cases<br>Intrapartum stillbirth (N = 193) n (%) | Controls<br>Livebirths (N = 381)<br>n (%) |
|---|---|---|---|
| **Sociodemographic factors** | | | |
| **Mean maternal age (yrs) (SD)** | 29.7 (5.8) | 29.9 (5.6) | 28.7 (5.4) |
| **Maternal age (yrs)** | | | |
| <20 | 6 (4.3) | 7 (4.1) | 16 (4.5) |
| 20–24 | 27 (19.3) | 26 (15.0) | 62 (17.4) |
| 25–29 | 41 (29.3) | 53 (30.6) | 133 (37.4) |
| 30–34 | 28 (20.0) | 47 (27.2) | 93 (26.1) |
| > = 35 | 38 (27.1) | 40 (23.1) | 52 (14.61) |
| Missing | 17 (10.8) | 20 (10.4) | 25 (6.6) |
| **Highest maternal education** | | | |
| No formal education | 0 (0.0) | 1 (0.5) | 2 (0.5) |
| Primary education | 2 (1.3) | 3 (1.6) | 5 (1.3) |
| Secondary education | 11 (7.0) | 10 (5.2) | 48 (12.6) |
| Undergraduate | 7 (4.5) | 18 (9.3) | 76 (20.0) |
| Any postgraduate | 4 (2.6) | 4 (2.1) | 7 (1.8) |
| Missing | 133 (84.7) | 157 (81.4) | 243 (63.8) |
| **Marital status** | | | |
| Married | 142 (90.5) | 159 (82.4) | 351 (92.1) |
| Not married | 6 (3.8) | 13 (6.7) | 9(2.4) |
| Missing | 9 (5.7) | 21 (10.9) | 21 (5.5) |
| **Employment status** | | | |
| Employed | 16 (10.2) | 18 (9.3) | 50 (13.1) |
| Self-employed | 51 (32.5) | 45 (23.3) | 108 (28.4) |
| Unemployed | 23 (14.7) | 18 (9.3) | 63 (16.5) |
| Missing | 67 (42.7) | 112 (58.0) | 160 (42) |
| **Religion** | | | |
| Christian | 149 (94.9) | 182 (94.3) | 360 (94.5) |
| Others | 4 (2.55) | 2 (1.0) | 5 (1.31) |
| Missing | 4 (2.55) | 9 (4.7) | 16 (4.20) |
| **Place of residence** | | | |
| Rural | 97 (61. 8) | 123 (63.7) | 221 (58.0) |
| Urban | 56 (35.7) | 64 (33.2) | 149 (39.1) |
| Missing | 4 (2.6) | 6 (3.1) | 11 (2.9) |
| **Ethnicity** | | | |
| Igbo | 151 (96.2) | 187 (96.9) | 372 (97.6) |
| Others | 5 (3,2) | 5 (2.6) | 5 (1.31) |
| Missing | 1 (0.64) | 1 (0.5) | 4 (1.1) |
| **obstetric and maternity care factors** | | | |
| **Parity** | | | |
| 0 | 60 (38.2) | 71 (36.8) | 128 (33.6) |
| 1–3 | 75 (47.8) | 89 (46.1) | 222 (58.3) |
| 4 or more | 18 (11.5) | 29 (15.0) | 28 (7.4) |
| Missing | 4 (2.6) | 4 (2.1) | 3 (0.8) |
| **Stillbirth history** | | | |
| No | 117(74.5) | 153(79.3) | 294 (77.2) |
| Yes | 13 (8.3) | 14 (7.3) | 29 (7.61) |
| Missing | 27 (17.2) | 26 (13.5) | 58 (15.22) |
| **Booking status** | | | |
| Booked in facility of childbirth | 62 (39.5) | 103 (53.4) | 309 (81.1) |
| Booked in different facility | 17 (10.8) | 16 (8.3) | 12 (3.2) |
| Not booked in a health facility | 78 (49.7) | 74 (38.3) | 60 (15.8) |
| **Number of antenatal visits** | | | |

*(Continued)*

**Table 2.** (Continued)

| | Cases<br>Antepartum stillbirth (N = 157) n (%) | Cases<br>Intrapartum stillbirth (N = 193) n (%) | Controls<br>Livebirths (N = 381)<br>n (%) |
|---|---|---|---|
| Unbooked | 78 (49.7) | 74 (38.3) | 60 (15.8) |
| 1–3 visits | 4 (2.6) | 9 (4.7) | 12 (3.15) |
| 4 or more visits | 57 (36.3) | 92 (47.7) | 262 (68.8) |
| Missing | 18 (11.5) | 18 (9.3) | 47 (12.3) |
| **Median gestational age weeks (SD)** | 38(4.2) | 38 (4.0) | 39 (1.4) |
| **Gestational age (weeks)** | | | |
| Preterm (28 –< 37 weeks) | 55 (35.0) | 53 (27.5) | 16 (4.2) |
| Term (37 weeks-40 weeks) | 88 (56.1) | 124 (64.3) | 338 (88.7) |
| Late term (41–42 weeks) | 12 (7.64) | 13 (6.7) | 27 (7.1) |
| Missing | 2(1.3) | 3 (1.6) | 0(0) |
| **Past obstetric complication** | | | |
| No | 143 (91.8) | 182 (94.3) | 351 (92.1) |
| Yes | 14 (8.9) | 11 (5.7) | 30 (5.7) |
| **Partogram use in labour** | | | |
| No | 105 (66.9) | 124 (64.3) | 253 (66.4) |
| Yes | 11 (7.0) | 17 (8.8) | 74 (19.4) |
| Not indicated | 41 (26.1) | 52 (26.9) | 54 (14.12) |
| **Mode of childbirth** | | | |
| Vaginal birth | 110 (70.1) | 106 (54.9) | 281 (73.8) |
| Caesarean Section | 47 (29.9) | 87 (45.1) | 100 (26.3) |
| **Referral status** | | | |
| Not referred | 83 (52.9) | 118( 61.1) | 332 (87.1) |
| Referred | 74 (47.1) | 75 (38.9) | 49 (12.9) |
| **Facility of childbirth** | | | |
| Public | 54 (34.4) | 44 (22.8) | 103 (27.0) |
| Private | 56 (35.7) | 68(35.2) | 147 (38.6) |
| Mission | 47 (29.9) | 81 (42.0) | 131 (34.3) |
| **Biological factors** | | | |
| **Medical comorbidities** | | | |
| No | 145 (92.4) | 187 (96.9) | 373 (97.9) |
| Yes | 12 (7.6) | 6 (3.1) | 8 (2.1) |
| **Obstetric complications in this pregnancy** | | | |
| No | 51/(32.5) | 48 (24.9) | 294 (77.2) |
| Yes | 106 (67.5) | 145 (75.1) | 87 (22.8) |
| **Fetal sex** | | | |
| Male | 86(54.8) | 103 (53.4) | 204 (53.5) |
| Female | 52 (33.1) | 72 (37.3) | 174 (45.7) |
| Missing | 19 (12.1) | 18 (9.3) | 3 (0.8) |
| **Birth weight for gestational age (GA)** | | | |
| Small for GA (<10th centile) | 8 (9.0) | 4(3.5) | 18 (7.3) |
| Normal for GA | 17 (19.1) | 27 (23.7) | 175 (71.1) |
| Large for GA (>90th centile) | 5 (5.6) | 13 (11.4) | 50 (20.3) |
| Missing | 59 (66.3) | 70 (61.4) | 3(1.22) |

The sociodemographic factor, maternal age, together with the obstetric and maternity care factors had good ability to discriminate between individuals with and without antepartum stillbirth (AUC 0.81), and this improved further to 0.85 on addition of the biological risk factors (Fig 3).

For ISB, after adjusting for maternal age, marital status and obstetric and maternity care factors, maternal age (aOR 1.05, 95%CI 1.00–1.10); preterm compared with term birth (aOR 13.07, 95%CI 5.97–28.61); being referred from another facility (aOR 3.42,

**Table 3. Risk factors (adjusted odds ratios) for antepartum stillbirths with progressive adjustments.**

| Variable | OR adjusted for sociodemographic factors (95% CI) (n = 519) | OR adjusted for Socio + obstetric and maternity care factors (95% CI) (n = 447) | OR adjusted for sociodemographic + obstetric and maternity care + biological factors (95% CI) (n = 447) |
|---|---|---|---|
| **Sociodemographic factors** | | | |
| **Maternal age per 1 year increase in age** | 1.03 (1.00–1.07) | 1.05 (1.00–1.11) | 1.03 (0.98–1.09) |
| **obstetric and maternity care factors** | | | |
| **Parity** | | | |
| 0 | | 2.02 (1.15–3.56) | 1.87 (1.04–3.36) |
| 1–3 | | 1 (ref) | 1 (ref) |
| 4 or more | | 1.36 (0.56–3.31) | 1.24 (0.47–3.26) |
| **Gestational age** | | | |
| Term | | 1 (ref) | 1(ref) |
| Late term | | 1.76 (0.74–4.20) | 1.58 (0.65–3.84) |
| Preterm | | 13.61 (6.24–29.65) | 14.29(6.31–32.38) |
| **Referral status** | | | |
| Not referred | | 1 (ref) | 1 (ref) |
| Referred | | 3.75 (1.96–7.17) | 3.15 (1.61–6.15) |
| **Number of ANC visits** | | | |
| Unbooked | | 2.58 (1.37–4.85) | 2.07 (1.07–4.00) |
| 1–3 visits | | 0.81 (0.18–3.55) | 0.94 (0.20–4.37) |
| 4 or more visits | | 1 (ref) | 1 (ref) |
| **Biological factors** | | | |
| **Medical comorbidities** | | | |
| No | | | 1 (ref) |
| Yes | | | 1.88 (0.55–6.41) |
| **Obstetric complication(s)** | | | |
| No | | | 1 (ref) |
| Yes | | | 4.04(2.35–6.94) |

95%CI 1.74–6.70); having a caesarean birth compared with vaginal birth (aOR 2.01, 95%CI 1.13–3.57); and not using partogram when indicated (aOR 2.72, 95%CI 1.23–6.02) remained statistically significant, but parity and antenatal visits were not significantly associated with the outcome (Table 4).

After adjusting for biological factors, similar to the antepartum model, maternal age was no longer significantly associated with ISB and the magnitude of association was marginally attenuated for the obstetric and maternity care factors of being preterm (aOR 11.28, 95%CI 4.66–27.24) and being referred (aOR 2.50, 95%CI 1.19–5.24), but the effect size of the aOR for not using partogram increased marginally (aOR 2.92, 95%CI 1.23–6.95) and caesarean section was no longer significantly associated with ISB. The effect of having any obstetric complication was much higher for ISB compared with ASB (aOR 10.71, 95%CI 5.92–19.37) (Table 4). There was no indication of multicollinearity in the model (mean VIF—1.44, maximum VIF– 2.35, $r_s$ = -0.6 to 0.4).

Maternal age and marital status (sociodemographic factors) together with the obstetric and maternity care factors had good ability to discriminate between individuals with and without ISB (AUC 0.80), and this increased to 0.87 on addition of biological factors (Fig 4).

The complete case analysis for antepartum and intrapartum models were comparable with the imputed models (S5 and S6 Tables), hence the complete case analysis models were used for interpretation. PAF estimates showed that within our population the risk factors for ASB with the greatest potential impact for risk reduction were obstetric complications (49%) and

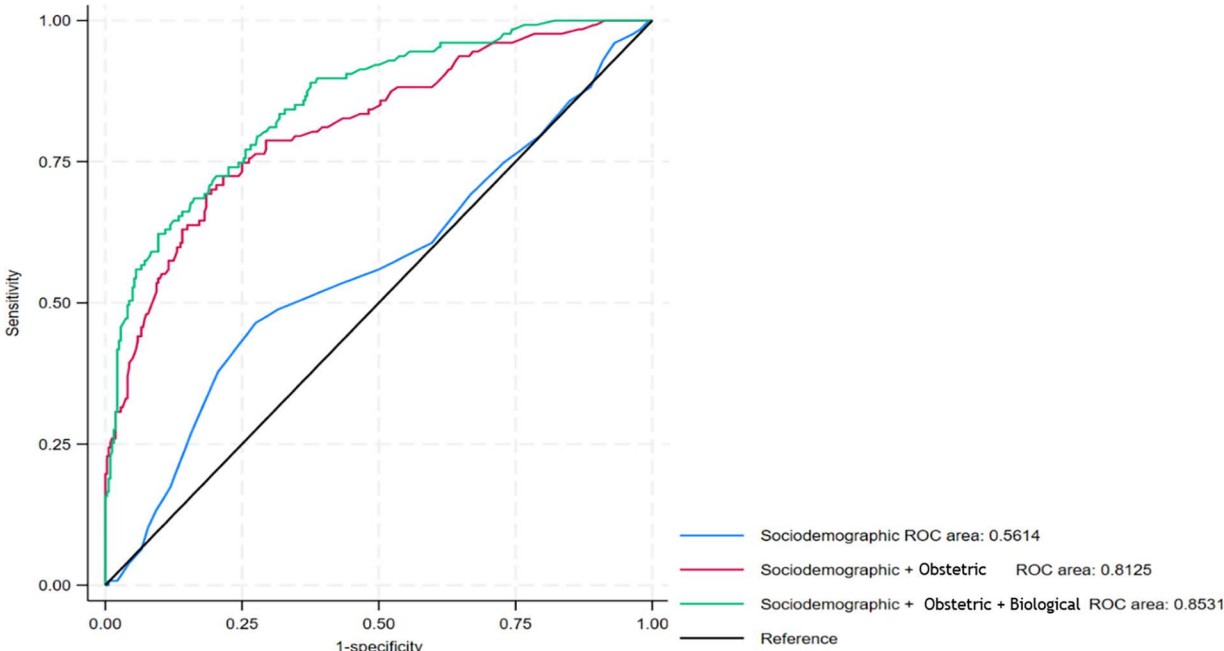

**Fig 3. Area under the receiver-operating characteristic (AUROC) curve for sociodemographic, obstetric and maternity care and biological factors for antepartum stillbirths.**

referral from another facility (34%), while for ISB, obstetric complications (70%) and failure to use a partogram when indicated (62%) were the risk factors with the greatest potential impact (Table 5).

The most common obstetric complications were Hypertensive disorders of pregnancy (HDP) (22.3%) and prolonged labour (23.8%) for ASB and ISB respectively (Table 6). The risk factors with the greatest elevation in the odds of stillbirth were HDP and Antepartum haemorrhage respectively. The leading obstetric complications with the greatest potential for impact were Hypertensive disorders of pregnancy (21%) for ASB and prolonged labour (20%) for ISB.

## Discussion

The overall stillbirth rate in the 20 hospitals was 38 per 1000 births, while the rates of antepartum stillbirths (ASB) and intrapartum stillbirths (ISB) were 16 and 19 per 1,000 births respectively. Four risk factors: nulliparity, preterm birth, being referred from another facility and having obstetric complication/s, were independently associated with both ASB and ISB. However, being unbooked for pregnancy care only increased the risk of ASB and not ISB, and maternal age was not associated with either type of stillbirths. Importantly not using a partogram when indicated significantly increased the risk of ISB.

We found a lower stillbirth rate in our study, about a third [7] and a fifth [6] of the rates reported in studies conducted in tertiary facilities in Imo state which was not surprising as tertiary hospitals care for high-risk women from peripheral facilities. ISB is a sensitive indicator of the timeliness and quality of intrapartum care [1]. About half of the stillbirths in our study occurred in the intrapartum period compared with 8% in Western Europe [1]. This highlights an opportunity for improvement using critical life-saving interventions in the intrapartum period. In comparison with other studies that informed our theoretical framework [15–17], neither maternal age nor marital status were associated with increased odds of

**Table 4. Risk factors for intrapartum stillbirths with progressive adjustments.**

| Variable | OR adjusted for Sociodemographic factors (95% CI) (n = 508) | OR adjusted for Sociodemographic + obstetric and maternity care factors (95% CI) (n = 450) | OR adjusted for sociodemographic + obstetric and maternity care + biological factors (95% CI) (n = 450) |
|---|---|---|---|
| **Sociodemographic factors** | | | |
| **Maternal age per 1 year increase in age** | 1.06 (1.02–1.11) | 1.05 (1.00–1.10) | 1.03 (0.98–1.09) |
| **Marital status** | | | |
| Married | 1 (ref) | 1 (ref) | 1 (ref) |
| Not married | 4.41 (1.74–11.18) | 2.54 (0.81–7.97) | 2.54 (0.65–9.92) |
| **Obstetric and maternity care factors** | | | |
| **Parity** | | | |
| 0 | | 1.70 (0.98–2.93) | 1.29 (0.71–2.37) |
| 1–3 | | 1 (ref) | 1 (ref) |
| 4 or more | | 1.97 (0.91–4.24) | 1.61 (0.67–3.87) |
| **Gestational age** | | | |
| Term | | 1 (ref) | 1 (ref) |
| Late term | | 1.28 (0.51–3.23) | 1.03 (0.38–2.75) |
| Preterm | | 13.07 (5.97–28.61) | 11.28 (4.66–27.24) |
| **Referral status** | | | |
| No | | 1 (ref) | 1 (ref) |
| Yes | | 3.42 (1.74–6.70) | 2.50 (1.19–5.24) |
| **Number of ANC visits** | | | |
| Unbooked | | 1.45 (0.74–2.84) | 1.28 (0.40–2.16) |
| 1–3 visits | | 1.67 (0.56–4.99) | 1.50 (0.43–5.04) |
| 4 or more visits | | 1 (ref) | 1 (ref) |
| **Mode of childbirth** | | | |
| Vaginal birth | | 1 (ref) | 1 (ref) |
| Caesarean birth | | 2.01 (1.13–3.57) | 0.68 (0.35–1.32) |
| **Partogram use** | | | |
| No, but indicated | | 2.72 (1.23–6.02) | 2.92 (1.23–6.95) |
| Yes | | 1 (ref) | 1 (ref) |
| Not indicated | | 2.57 (0.97–6.74) | 3.81 (1.33–10.92) |
| **Biological factors** | | | |
| **Obstetric complication(s)** | | | |
| No | | | 1 (ref) |
| Yes | | | 10.71 (5.92–19.37) |

ASB or ISB in our population. This could be due to the limited variability of these factors in our study population. Preterm birth and referral from another health facility were associated with increased odds of both stillbirth types. In our study, preterm birth is likely to be an indicator of non-viability or poor development of the fetus leading to antepartum or intrapartum

**Table 5. Population attributable fractions for risk factors in final model.**

| Risk factors | Antepartum stillbirths | | Intrapartum stillbirths | |
|---|---|---|---|---|
| | PAF (%) | 95% CI | PAF (%) | 95% CI |
| Prematurity | 32.9 | 27.7–37.7 | 26.5 | 20.3–32.3 |
| Nulliparity | 21.5 | 2.4–36.9 | - | - |
| Unbooked pregnancy | 27.5 | 6.6–43.7 | - | - |
| Referral from another hospital | 34.4 | 22.8–44.3 | 25.2 | 11.6–36.7 |
| Not using a partogram when indicated | - | - | 62.2 | 27.0–80.5 |
| Any pregnancy complication | 49.2 | 39.6–57.2 | 70.0 | 65.4–74.0 |

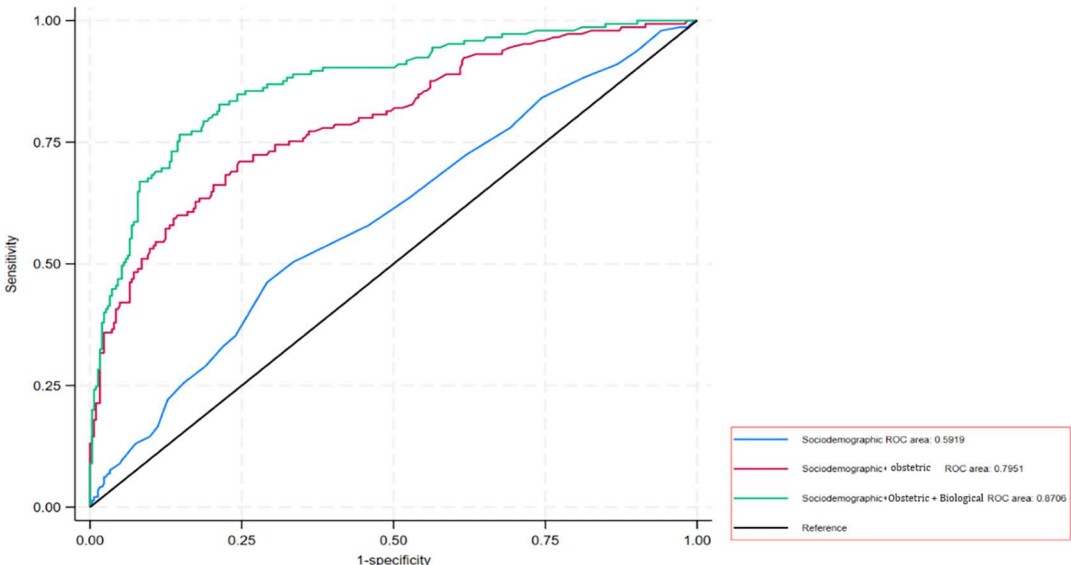

**Fig 4. Area under the receiver-operating characteristic (AUROC) curve for sociodemographic, obstetric and maternity care and biological factors for intrapartum stillbirths.**

death before reaching term. This can be caused by complications of hypertension, diabetes, anaemia among others, which were common in the study population and reported by other studies [6–8,18–22]. Thus, improving detection and care for these complications is essential. Women who were referred from a different hospital emerged as a particularly vulnerable group for both types of stillbirths with over four-fold and three-fold increase in risk for ASB and ISB respectively. Previous studies found that when high-risk pregnancies needed referral, longer travel distances, lack of ambulances for easy transfer, and issues such as lack of beds or limited capacity at receiving hospitals further necessitating second referrals increased the risk of stillbirth [8,16]. These challenges are not unique to Nigeria and were reported in a systematic reviews including other countries in Africa [23] and another review from India

**Table 6. Effect of obstetric complications on stillbirths after adjusting for sociodemographic and intermediate factors.**

| Complication | Antepartum stillbirths (n = 157) | | | | Intrapartum stillbirths (n = 193) | | | |
|---|---|---|---|---|---|---|---|---|
| | n (%) | Crude OR (95% CI) | AOR (95% CI) | PAF (95% CI) | n (%) | Crude OR | AOR | PAF (95% CI) |
| Prolonged labour | - | - | - | - | 46 (23.8) | 2.74(1.71–4.38) | 6.85(3.23–14.50) | 20% (17.4–22.5) |
| Malaria | 5 (3.2) | 2.47 (0.71–8.67) | 1.34 (0.22–8.18) | 0.8% (-3.4–4.9) | 9 (4.6) | 3.68 (1.22–11.13) | 4.73 (1.17–19.16) | 3.8% (2.4–5.2) |
| Premature Rupture of membrane | 16 (10.2) | 3.82 (1.73–8.43) | 1.13 (0.23–5.51) | 0.9% (-10.8–11.3) | 22 (11.4) | 4.33 (2.05–9.13) | 4.84 (1.46–16.06) | 9.8% (6.7–12.9) |
| Abnormal presentation | 14 (8.9) | 7.36 (2.60–20.81) | 7.80 (2.20–27.67) | 6.9% (5.6–8.1) | 25 (13.0) | 11.19 (4.21–29.73) | 17.14 (5.03–58.38) | 13.0% (12.0–14.0) |
| Uterine rupture | 4(2.5) | 3.29 (0.73–14.89) | 2.84 (0.43–18.98) | 6.9% (5.6–8.1) | 13 (6.7) | 9.1 (2.56–32.33) | 8.32 (1.75–39.57) | 1.5% (- 0.06–3.1) |
| Antepartum Haemorrhage | 15(9.6) | 3.63 (1.12–11.73) | 5.85 (1.74–19.60) | 7.8% (5.9–9.8) | 31 (16.1) | 7.91 (3.68–16.99) | 17.75 (5.60–56.27) | 15.6% (14.5–16.7) |
| Hypertensive disorders of Preg. | 35 (22.3) | 7.52 (3.92–14.45) | 14.20(5.76–34.99) | 21.3% ( | 23(11.9) | 3.54 (1.78–10.06) | 8.21 (3.08–21.89) | 12.1% (10.4–13.7) |

[24]. Sometimes, women chose to skip closer health facilities due to financial barriers or past negative experiences with care [16,25]. Although women generally require referral when they develop an obstetric or fetal complication which may account for the excess risk of stillbirth associated with referral, there is also a need to explore and understand possible additional risks within the referral mechanism itself.

While previous studies have found associations between women's parity and booking status with stillbirths in general [15,17], in our study we found that these two factors were only important for antepartum but not intrapartum stillbirths. Unbooked women had twice the risk of ASB compared with booked women who had at least four antenatal visits. We found that nulliparous individuals had nearly twice the risk of ASB compared with women with 1–3 previous births. These findings suggest that early antenatal booking and tailored care for nulliparous women may help detect risk factors and emerging complications to reduce the risk of ASB. In contrast, a systematic review and metanalysis from sub-saharan Africa [26] and specific study in Ethiopia [27] showed that nulliparity was protective for stillbirth. It is worth investigating further into reasons for the increased stillbirth risk for nulliparous women in Nigeria in comparison with other African countries. In cases of ISB, when a woman arrives at a health facility with a fetal heart rate present, the fetal survival depends upon the swiftness and quality of care the woman receives during labour. Our study suggests that using a partogram when indicated has the potential to eliminate about 62% of ISB. Visually tracking labour progress using a partogram and monitoring key indicators enables timely intervention for complications. The overall use of partogram in our study population was poor and in a recent survey of obstetric care providers in Imo state, although 94% claimed to use partograms for monitoring in labour, only 2% of sampled medical records contained information about its use [28]. Factors contributing to this inconsistency included time constraints, staff shortages, unavailability of partograms, and insufficient training [28].

The presence of obstetric complications was an important risk factor for both types of stillbirths. PAF estimates suggest that 49% of ASB and 70% of ISB in our population could potentially be eliminated if these complications were absent. Hypertensive disorders of pregnancy are a leading cause of stillbirth in this region observed in 7% - 14% of stillbirths [6–8,18]. In our study, when these disorders were present, they resulted in an eleven-fold and four-fold increase in the odds of ASB and ISB respectively. Abnormal fetal presentation was associated with more than five-fold increased odds of ASB and over an eight-fold increase in the odds of ISB. Although, we did not find a significant association between caesarean section and stillbirth, some women with abnormal fetal presentation need caesarean section which has been shown to have sociocultural implications leading to delayed consent [29]. Malaria in pregnancy has been linked to an increased risk of stillbirth but was only associated with an increased risk in ISB in our study, potentially due to the small number of women with a documented diagnosis of malaria. The AUROC curve confirms that collectively nulliparity, preterm birth, unbooked pregnancy, referral, failure to use partogram and pregnancy complications explained the risk of stillbirth in this population. This emphasises the need for high quality antenatal and, intrapartum care, and CEmOC as crucial interventions for risk reduction.

The strengths of our study lie in this being the first of its kind to assess and compare risk factors for ASB and ISB in secondary facilities in Imo state. We also used a robust case-control study design. However, we acknowledge important limitations in our study. Our hospital selection was limited to LGAs considered safe at the point of data collection introducing potential selection bias and affecting the generalisability of our findings. *A consequence of the potential selection bias would be an underestimation or overestimation of stillbirth incidence.* Secondly, important variables such as fetal weight for gestational age, socioeconomic status, presence of fetal distress amongst others were excluded from our analysis due to large

proportions of missing data, thus, we cannot completely rule out the possibility of residual confounding. Another limitation is that stillbirths types in hospital records are prone to misclassification as have been shown in previous studies [8,30]. These limitations suggest that while our study offers valuable insights, the results should be interpreted with caution. Future research should aim for more rigorous designs, to address these limitations. This study serves as a baseline on which other hypothesis driven studies could be built. Despite these constraints, our findings provide valuable insights into the primary risk factors for the two types of stillbirths, offering a foundation for targeted interventions and also emphasising the necessity for improved clinical care documentation to aid data collection for future research.

Our findings have significant policy and planning implications. It is imperative to establish robust systems for classifying and reporting stillbirths. Improving data completeness and reporting requires periodic training for health workers and strengthening data collection systems infrastructure to ensure accurate and consistent reporting. A shift to digital health records could help streamline data collection, potentially reduce errors and improve timely access to stillbirth data. However, the implementation of electronic medical records in Nigeria has been fraught with many challenges, including poor infrastructure, funding constraints and resistance by health workers [31]. Progress will remain slow without relevant data in the region. Equally crucial is the need to strengthen referral pathways and emergency readiness in secondary hospitals. Several interventions have been implemented for stillbirth reduction in the region [32,33], implementing interventions targeting nulliparous women, routine screening for hypertensive disorders during antenatal care, mandatory use of partograms for labour monitoring, and the introduction of early counselling services for expectant mothers with abnormal presentations are required. These will improve care for high-risk groups, and proactively mitigate/reduce potential risk factors, with the goal of reducing stillbirth rates and improving outcomes for both mothers and babies.

## Supporting information

**S1 Table. Sample size calculation using Kelsey equation.**
(PDF)

**S2 Table. Design effect and missing data inflation calculation.**
(PDF)

**S3 Table. Health facility included in the study and their characteristics.**
(PDF)

**S4 Table. Univariable analysis of risk factors associated with antepartum and intrapartum Stillbirths.**
(PDF)

**S5 Table. Comparison between complete case and multiple imputation models for antepartum stillbirths.**
(PDF)

**S6 Table. Comparison between complete case and multiple imputation models for intrapartum stillbirths.**
(PDF)

**S7 Table. Effect of obstetric complications on stillbirths after adjusting for sociodemographic and intermediate factors.**
(PDF)

**S1 Fig. Flowchart for selection of study sample included in the analysis.**
(PDF)

## Author contributions

**Conceptualization:** Uchenna Gwacham-Anisiobi, Charles Opondo, Jennifer J. Kurinczuk, Manisha Nair.

**Data curation:** Uchenna Gwacham-Anisiobi, Geoffrey Anyaegbu.

**Formal analysis:** Uchenna Gwacham-Anisiobi, Charles Opondo, Tuck Seng Cheng.

**Funding acquisition:** Uchenna Gwacham-Anisiobi.

**Investigation:** Uchenna Gwacham-Anisiobi.

**Methodology:** Uchenna Gwacham-Anisiobi, Jennifer J. Kurinczuk, Manisha Nair.

**Project administration:** Uchenna Gwacham-Anisiobi, Manisha Nair.

**Resources:** Uchenna Gwacham-Anisiobi, Geoffrey Anyaegbu, Manisha Nair.

**Supervision:** Charles Opondo, Jennifer J. Kurinczuk, Manisha Nair.

**Validation:** Uchenna Gwacham-Anisiobi, Tuck Seng Cheng, Geoffrey Anyaegbu.

**Visualization:** Uchenna Gwacham-Anisiobi.

**Writing – original draft:** Uchenna Gwacham-Anisiobi.

**Writing – review & editing:** Uchenna Gwacham-Anisiobi, Charles Opondo, Tuck Seng Cheng, Jennifer J. Kurinczuk, Geoffrey Anyaegbu, Manisha Nair.

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
