## [Decision Letter · Decision Letter 0]

29 May 2024

PGPH-D-24-00500

Rates and risk factors for antepartum and intrapartum stillbirths in 20 secondary hospitals in Imo state, Nigeria: a hospital-based case control study.

Dear Dr. Gwacham-Anisiobi,

Thank you for submitting your manuscript to PLOS Global Public Health. After careful consideration, we feel that it has merit but does not fully meet PLOS Global Public Health’s publication criteria as it currently stands. Therefore, we invite you to submit a revised version of the manuscript that addresses the points raised during the review process.

Please note that we have only been able to secure a single reviewer to assess your manuscript. We are issuing a decision on your manuscript at this point to prevent further delays in the evaluation of your manuscript. Please be aware that the editor who handles your revised manuscript might find it necessary to invite additional reviewers to assess this work once the revised manuscript is submitted. However, we will aim to proceed on the basis of this single review if possible.

Please see the comments form the reviewer in the attached document. Could you please revise the manuscript to carefully address the concerns raised?

We look forward to receiving your revised manuscript.

Kind regards,

Steve Zimmerman, PhD

PLOS Staff Editor

Journal Requirements:

3. Please provide separate figure files in .eps or .eps format only and remove any figures embedded in your manuscript file. Please also ensure all files are under our size limit of 10MB.

4. We have noticed that you have uploaded Supporting Information files, but you have not included a list of legends. Please add a full list of legends for your Supporting Information files after the references list.

5. Some material included in your submission may be copyrighted. According to PLOS’s copyright policy, authors who use figures or other material (e.g., graphics, clipart, maps) from another author or copyright holder must demonstrate or obtain permission to publish this material under the Creative Commons Attribution 4.0 International (CC BY 4.0) License used by PLOS journals. Please closely review the details of PLOS’s copyright requirements here: PLOS Licenses and Copyright. If you need to request permissions from a copyright holder, you may use PLOS's Copyright Content Permission form.

Potential Copyright Issues:

Fig 2: please (a) provide a direct link to the base layer of the map (i.e., the country or region border shape) and ensure this is also included in the figure legend; and (b) provide a link to the terms of use / license information for the base layer image or shapefile. We cannot publish proprietary or copyrighted maps (e.g. Google Maps, Mapquest) and the terms of use for your map base layer must be compatible with our CC-BY 4.0 license.

* U.S. Geological Survey (USGS) - All maps are in the public domain. (http://www.usgs.gov )

* PlaniGlobe - All maps are published under a Creative Commons license so please cite “PlaniGlobe, http://www.planiglobe.com , CC BY 2.0” in the image credit after the caption. (http://www.planiglobe.com /?lang=enl)

* Natural Earth - All maps are public domain. (http://www.naturalearthdata.com/about/terms-of-use/ )

Additional Editor Comments (if provided):

Reviewers' comments:

Reviewer's Responses to Questions

**Comments to the Author**

1. Does this manuscript meet PLOS Global Public Health’s publication criteria ? Is the manuscript technically sound, and do the data support the conclusions? The manuscript must describe methodologically and ethically rigorous research with conclusions that are appropriately drawn based on the data presented.

Reviewer #1: Yes

2. Has the statistical analysis been performed appropriately and rigorously?

Reviewer #1: I don't know

3. Have the authors made all data underlying the findings in their manuscript fully available (please refer to the Data Availability Statement at the start of the manuscript PDF file)?

Reviewer #1: No

4. Is the manuscript presented in an intelligible fashion and written in standard English?

Reviewer #1: Yes

5. Review Comments to the Author

Reviewer #1: The data availability statement did not indicate all data are included but has shared some and others are on the manuscript. The set of question asks for inclusion of all data and as such my response was no

6. PLOS authors have the option to publish the peer review history of their article (what does this mean? ). If published, this will include your full peer review and any attached files.

**Do you want your identity to be public for this peer review?** For information about this choice, including consent withdrawal, please see our Privacy Policy .

Reviewer #1: **Yes: ** Nebreed Fesseha(MD)

---

## [Decision Letter · Decision Letter 1]

9 Sep 2024

Rates and risk factors for antepartum and intrapartum stillbirths in 20 secondary hospitals in Imo state, Nigeria: a hospital-based case control study.

PGPH-D-24-00500R1

Dear Dr Gwacham-Anisiobi,

We are pleased to inform you that your manuscript 'Rates and risk factors for antepartum and intrapartum stillbirths in 20 secondary hospitals in Imo state, Nigeria: a hospital-based case control study.' has been provisionally accepted for publication in PLOS Global Public Health.

If your institution or institutions have a press office, please notify them about your upcoming paper to help maximize its impact. If they'll be preparing press materials, please inform our press team as soon as possible -- no later than 48 hours after receiving the formal acceptance. Your manuscript will remain under strict press embargo until 2 pm Eastern Time on the date of publication. For more information, please contact globalpubhealth@plos.org .

Best regards,

Nazmul Alam, MPH, DrPH

Academic Editor

Reviewer Comments (if any, and for reference):

Reviewer's Responses to Questions

**Comments to the Author**

1. If the authors have adequately addressed your comments raised in a previous round of review and you feel that this manuscript is now acceptable for publication, you may indicate that here to bypass the “Comments to the Author” section, enter your conflict of interest statement in the “Confidential to Editor” section, and submit your "Accept" recommendation.

Reviewer #1: All comments have been addressed

Reviewer #2: All comments have been addressed

2. Does this manuscript meet PLOS Global Public Health’s publication criteria ? Is the manuscript technically sound, and do the data support the conclusions? The manuscript must describe methodologically and ethically rigorous research with conclusions that are appropriately drawn based on the data presented.

Reviewer #1: Yes

Reviewer #2: Yes

3. Has the statistical analysis been performed appropriately and rigorously?

Reviewer #1: I don't know

Reviewer #2: Yes

4. Have the authors made all data underlying the findings in their manuscript fully available (please refer to the Data Availability Statement at the start of the manuscript PDF file)?

Reviewer #1: Yes

Reviewer #2: No

5. Is the manuscript presented in an intelligible fashion and written in standard English?

Reviewer #1: Yes

Reviewer #2: Yes

6. Review Comments to the Author

Reviewer #1: All comments are well addressed

Reviewer #2: This paper highlights very important findings on stillbirths in Imo State Nigeria and important socioeconomic, maternal and biological factors affecting still births that add to the scientific and academic body of work on maternal and neonatal care.

From the responses provided, the authors have addressed all comments raised by previous reviewers. Some responses and changes made to the manuscript include the following:

1. Provided clarity on consent and ethical approval received.

2. Expanded on the limitations of the studies - majorly around security challenges affecting selection of study sites.

3. Included additional references from other geographies comparing results.

4. Provided more context on the missing data and it's impact on interpretation of results.

5. Explained the inability to analyze the association of the mode of childbirth (SVD, assisted vaginal birth, elective and emergency CS) with intrapartum stillbirth due to few cases of assisted vaginal births and elective CS.

6. Included definition of unclassified and how it differs from the standard coding of deaths.

7. Expanded on the recommendations for strengthening documentation and data systems as a result of the incomplete or missing data sets found.

8. Provided the areas and approaches for follow up research based on the limitations of their study.

However, there is still the issue of public accessibility of the data based on PLOS journal guidelines. The authors have indicated that not all data seta can be made publicly available until the publication of some work. I'll defer to Editor on the level of flexibility that can be granted.

Barring any concerns from the first reviewers and if the issue of access of their data is addressed, herby I recommend for publication.

thanks,

Dr Oluwaseun Aladesanmi

7. PLOS authors have the option to publish the peer review history of their article (what does this mean? ). If published, this will include your full peer review and any attached files.

**Do you want your identity to be public for this peer review?** For information about this choice, including consent withdrawal, please see our Privacy Policy .

Reviewer #1: **Yes: ** Nebreed Fesseha(MD)

Reviewer #2: **Yes: ** Dr. Oluwaseun Aladesanmi
